# Evidence-based complementary and alternative medicine conventional surgery combined with traditional Chinese medicinal retention enema for tubal obstructive infertility: A systematic review and meta-analysis

**Sijia Xu[1,2‡], Shuo Jin[2‡], Liuqing Yang[2], Ling Wang[2], Qin Zhang[2] \***

1 The First Affiliated Hospital of Zhejiang Chinese Medical University (Zhejiang Provincial Hospital of Chinese Medicine), Zhejiang, China, 2 Hangzhou Traditional Chinese Medicine Hospital, Zhejiang Chinese Medicine University, Zhejiang, China

‡ SX and SJ are co-first authors on this work.
* zhqin@zjwh.gov.cn

**Data Availability Statement:** All relevant data are within the paper and Supporting Information files.

## Abstract

### Background

Chinese medicinal retention enemas have gradually attracted the attention of clinicians as an alternative approach for tubal obstructive infertility. The purpose of this study was to investigate the efficacy and safety of conventional surgery combined with traditional Chinese medicinal retention enemas for the treatment of tubal obstructive infertility.

### Materials and methods

Eight electronic databases were searched from their inception to November 30, 2022. To assess the efficacy and safety of different treatments, following outcomes were measured: clinical pregnancy rate, clinical total effective rate, incidence of ectopic pregnancy, the improvement of Traditional Chinese Medicinal (TCM) symptoms, the improvement of the signs of obstructive tubal infertility and side effects.

### Results

A total of 23 Randomized Controlled Trials (RCTs) with 1909 patients met the inclusion criteria. The pooled results showed a higher pregnancy rate in the experimental group than in the control group (RR 1.75, 95% CI [1.58, 1.94], Z = 10.55, $P<0.00001$). The clinical total effective rate in the experimental group was higher than that in the control group (RR 1.28, 95% CI [1.23, 1.34], Z = 11.07, $P<0.00001$). The incidence of ectopic pregnancy in the experimental group was lower than that in the control group (RR 0.40, 95% CI [0.20, 0.77], Z = -2.73, $P = 0.01$).

**Funding:** This research was financially supported by the Construction Program for National Famous Traditional Chinese Medicine Experts Inheritance Studio in 2022 (No.75 [2022] of Chinese Medicine People's Education Letter).The funders had no role in study design, data collection and analysis, decision to publish, or preparation of the manuscript.

**Competing interests:** The authors have declared that no competing interests exist.

## Conclusion

Based on current evidence, we concluded that conventional surgery combined with traditional Chinese medicinal retention enema for tubal obstructive infertility was superior to conventional surgery alone in improving the clinical pregnancy rate, improving clinical total effective rate, improving TCM symptoms, improving the signs of obstructive tubal infertility and lowering the incidence of ectopic pregnancy. However, further clinical trials with high-quality methodologies need to be conducted.

## Introduction

Tubal infertility is one of the main causes of female infertility, accounting for 25–35% of infertility [1]. The global rate of infertility in women is approximately 15.5%, and the incidence is increasing. In a study with 3018 infertility patients in China, we found that 23.46% of the patients had infertility due to tubal factors, especially in people over 35 years old, which is the leading cause of infertility. Tubal Obstructive Infertility refers to infertility caused by damage to the function and structure of the fallopian tubes due to pelvic infection, gynecological surgery, and other factors, causing chronic inflammatory damage and tissue fibrosis, resulting in inflammatory changes in the fallopian tubes, with pathological changes such as swelling, exudation, fluid accumulation, thickening, adhesion, stiffness, distortion, occlusion, and inaccessibility, leading to obstruction of sperm-egg union or delivery. Tube obstruction can lead to reduced fertility among affected women, accounting for 25 to ~ 35% of all causes of infertility. Pelvic inflammatory diseases, ectopic pregnancy, history of surgery in the uterine cavity, pelvic and abdominal operations, appendicitis, and endometriosis were found to be significant risk factors for tubal infertility [2, 3]. Routine therapy for tubal infertility includes hydrotubation, gynaecological endoscopic operation and interventional recanalization [4]. Although the operation can treat adhesion of fallopian tubes and restore anatomical morphology, inflammatory injury and endothelial circulatory disturbances still remain. These treatments do not recover physiological functions of the fallopian tube. Thus, many problems still exist, such as the high recurrence rate of oviduct inflammation, ectopic pregnancy soon after the operation [5]. Follow-up treatments are essential.

Traditional Chinese medicine (TCM) is a complete healing system developed in China about 3000 years ago, including herbal medicine, acupuncture, moxibustion, etc. In addition to acupuncture, TCM treatment for tubal infertility includes administration of Chinese medicine through various routes including oral administration, enema, local application, iontophoresis, and intrauterine injection [6]. Among these, traditional Chinese medicinal retention enema is one of the characteristic therapeutic methods of TCM, with a long-standing history and simple and significant clinical effect. However, current evidence of conventional surgery combined with TCM medicinal retention enema remains unsatisfactory. Therefore, a meta-analysis of randomized controlled trials (RCTs) was conducted in this work to evaluate the efficacy and safety of TCM. We conducted a new systematic review to evaluate the efficiency and safety of conventional surgery combined with TCM retention enema in the treatment of tubal obstructive infertility.

## Materials and methods

### Search strategy

The protocol has been registered in PROSPERO (ID: CRD42022325825). We conducted a systematic search for relevant documents in the Chinese and English databases, and the search

was conducted between the inception of each database and February 25, 2022. The following eight databases are included: PubMed, EMBASE, Web of Science, The Cochrane Library, Chinese Biomedical Literature Database (CBM), Chinese National Knowledge Infrastructure (CNKI), Chinese Science and Technology Periodical Database (VIP), Wanfang Database. There were no language restrictions for publications. Various synonyms of the concepts of "fallopian tube diseases", "Chinese medicine", "enema" and "randomized controlled trials" were combined by "And" to construct the search strategies.

## Inclusion and exclusion criteria

Inclusion criteria were as follows:

(1) Types of studies. We will include only randomized controlled trials published in both Chinese and English regardless of blinding and allocation concealment.

(2) Types of participants. All participants who have been diagnosed with fallopian tubal occlusion of female infertility [7] will be included. There are no restrictions on age, region, nationality, religion, ethnicity, sources, or courses of disease.

(3) Types of interventions. There was no requirement for the intervention course, and the specific requirements of the control group and the experimental group are as follows. The control group received only conventional operation for tubal obstructive infertility. All patients enrolled in the study received conventional operation for tubal obstructive infertility, including hydrotubation, gynaecological endoscopic operation, and interventional recanalization. The treatment group was supplemented with traditional Chinese medicinal retention enemas in addition to the conventional operation.

The exclusion criteria were as follows:

(1) Animal experiments, reviews, case reports, and non-randomized controlled trials were excluded.

(2) Acute and subacute inflammatory reactions of the genitals, luteal phase defects, ovulation disorders, serious cardiovascular and cerebrovascular diseases, and congenital or acquired defects will not be included. We excluded those with sexually transmitted diseases, previous abdominal or pelvic surgery history, and other gynecological diseases of the uterus and pelvis, such as endometriosis. Those with a combination of severe hemorrhoids, anal fissures and other contraindications to enemas; patients with more serious medical complications and psychiatric disorders, combined with other infertility factors.

(3) The intervention group did not use conventional surgery alone, or the experimental group was not given conventional operation with traditional Chinese medicine retention enema alone.

## Outcomes

The main evaluation indices were as follows:

(1) Clinical pregnancy rate: Intrauterine pregnancy should be identified with intrauterine gestational sac or foetal heartbeat by ultrasound.

(2) Clinical total effective rate: Markedly effective: Patients underwent intrauterine pregnancy or both sides of fallopian tube passable when evaluated with uterotubography. Effective: Oviduct obstruction showed some improvement when evaluated with uterotubography,

but patients had no pregnancy. Non-effect: The patency of the fallopian tube was unchanged from the previous.

Clinical total effective rate = ( number of markedly effective cases+number of effective cases)÷total number of cases

(3) Incidence of ectopic pregnancy.

(4) Side Effects.

## Data extraction

Two investigators (Sijia Xu and Shuo Jin) independently extracted and cross-checked the data. In cases of any disagreements, consensus was reached through discussions with the third investigator (Liuqing Yang). Data extraction information included the titles of the article, the first author, years of publication, setting, participants, sample size, treatment duration, intervention details, dropouts, outcome measures and adverse events. If data were incomplete or appeared incorrect, attempts were made to contact the authors to request additional information or clarify the data.

## Quality assessment

Two investigators assessed the methodological quality of the included studies independently using the risk of bias tools according to the Cochrane Handbook version 5.1.0 [8]. Disagreements were resolved through discussion with another investigator. Each item was assessed as low, high or unclear risk, with reasons recorded to support the judgements. RevMan 5.4, provided by the Cochrane Collaboration, was used to create plots demonstrating the risks of bias.

## Statistical analysis

This study involves bicategorical. The relative risk (RR) is used as a measure of effect for the bicategorical variables, and the software is able to obtain the point estimates and the 95% confidence interval (CI) for the 2. $I^2$ is an important index for making the heterogeneity judgement. Subgroup analysis by approach of surgery (hysteroscopy hydrotubation,laparoscopy, hysteroscopy laparoscopy and conventional hydrotubation), insertion depth of anal canal,follow-up time and TCM retention enema'time were performed if data permitted. If $I^2 < 50\%$, a fixed effects model was used; if $I^2 \geq 50\%$, a random effects model was used. For each combined analysis, the test of heterogeneity is measured using the cardinality statistic.

# Results

## Search results

Initially, we obtained 1043 articles through database search and manual search. These articles were screened by various methods and 23 eligible articles were finally included in this meta-analysis. The review process is as follows: after scanning titles and reviewing abstracts, 965 records were excluded with reasons of duplications, animal experiments, reviews, animal studies, or irrelevant studies. From the remaining 78 full papers, we further excluded 55 literatures for following reasons: intervention included other medical therapies(9); no clinical data for extraction (9); not RCT (3); poor thesis quality (4); data duplication (3).A flow diagram of the screening process is shown in Fig 1.

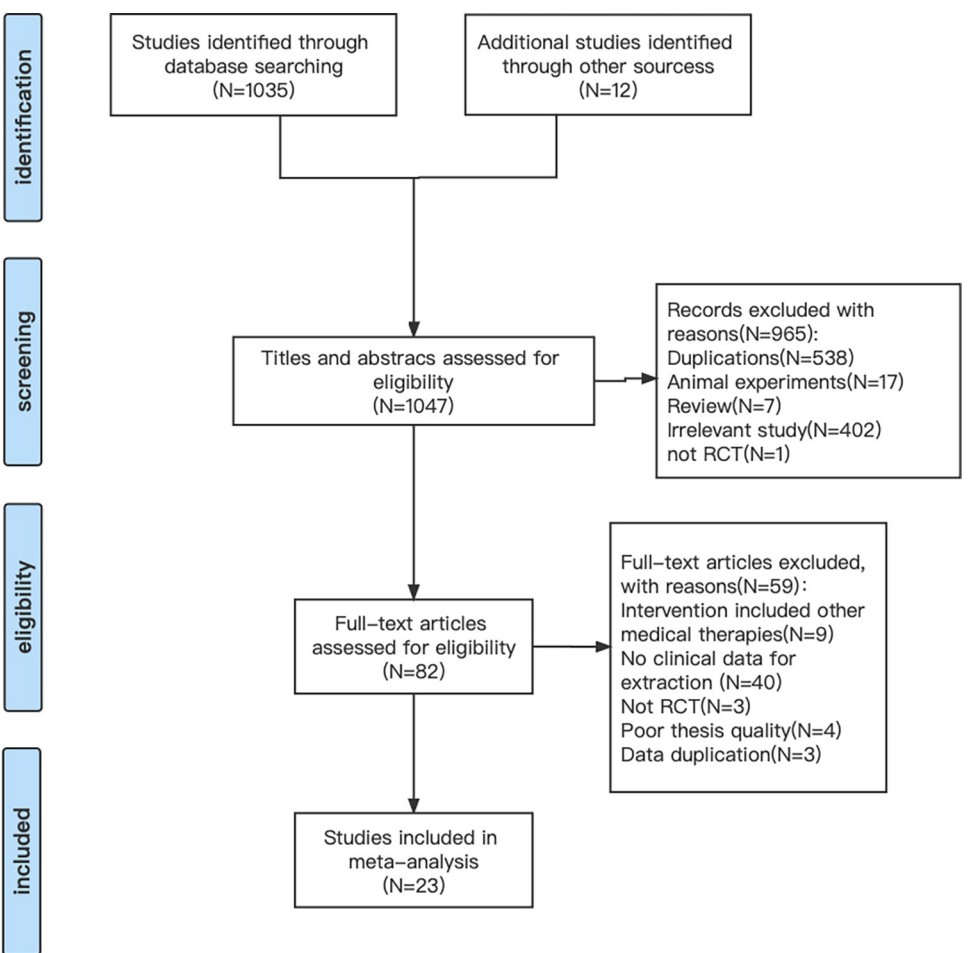

**Fig 1. Flow diagram for selected trials.** RCT, randomized controlled trial.

## Study characteristics

Table 1 presents the characteristics of the data extracted from the 23 articles included in the meta-analysis. All of them were conducted in China and were published in Chinese. The 23 studies incorporated into the meta-analysis involved a total of 1909 patients accurately diagnosed with tubal obstructive infertility (967 in the intervention group and 942 in the control group). The mean age ranged between 25.9 and 33.3 years, and the disease course ranged between 1.8and 6.21 years. The duration of treatment varied from 7 days to 3 months. The outcomes included were as follows: pregnancy rate and clinical total effective rate were used in all studies [9–31], incidence of ectopic pregnancy was used in 3 studies [21, 27, 29].

NM = not mention

## Risk of bias

The Cochrane Handbook 5.0.2 bias risk assessment tool was employed to evaluate the bias risk of all studies included in this meta-analysis. The evaluation results were input into Review Manager 5.4 software to generate a bias risk map. Fifteen studies [10, 15, 19, 21, 23, 26, 29] used random number tables or computers, indicating low risk, and 1 study [17] using treatment order were judged to be high risk. One studies [31] described specific methods of

**Table 1. Main characteristics of studies included in the review.**

| References | No. of patients (T/C) | Mean age (year) | | Disease course (year) | | Intervention | | Course | Follow-up time | TCM retention enema' time | Insertio-n depth | Outcomes |
|---|---|---|---|---|---|---|---|---|---|---|---|---|
| | | T | C | T | C | T | C | | | | | |
| Ling and Luo [9] | 34/34 | 30.3±3.2 | | 2.3±0.2 | | Conventional Hydrotubation +TCM retention enema | Conventional Hydrotubation | 20–30 days | 1 year | more than 2 hours | 15-20cm | Pregnancy rate, Clinical total effective rate |
| Zhang and Zhu [10] | 42/41 | 29.3 ±3.3 | 29.1 ±3.2 | 3.9 ±1.7 | 3.8 ±1.6 | Conventional Hydrotubation +TCM retention enema | Conventional Hydrotubation | 3 menstrual cycles | 1 year | more than 2 hours | 15–20 cm | Pregnancy rate, Clinical total effective rate |
| Ji [11] | 35/35 | 29.5 ±1.6 | 28.9 ±1.5 | 3.3 ±0.2 | 3.6 ±0.3 | Conventional Hydrotubation +TCM retention enema | Conventional Hydrotubation | 21 days | 1 year | 30minuts | NM | Pregnancy rate, Clinical total effective rate |
| Zhao [12] | 60/58 | 29.85 ±8.86 | 30.22 ±7.53 | 3.42 ±2.36 | 3.32 ±2.25 | Conventional Hydrotubation +TCM retention enema | Conventional Hydrotubation | 3 months | 6 months | NM | NM | Pregnancy rate, Clinical total effective rate |
| Tong et al. [13] | 47/46 | 29.7 ±2.3 | 28.9 ±2.4 | NM | NM | Laparoscopy and Conventional Hydrotubation +TCM retention enema | Laparoscopy | 1.5 months | 6 months —1 year | NM | NM | Pregnancy rate, Clinical total effective rate |
| Chen [14] | 24/24 | 31.2 ±4.0 | 30.5 ±3.8 | 3.0 ±0.9 | 2.6 ±0.6 | Hysteroscopy Laparoscopy+TCM retention enema | Hysteroscopy Laparoscopy | 3 months | 1 year | NM | NM | Pregnancy rate, Clinical total effective rate |
| Zhang and Hou [15] | 46/46 | 29.36 ±3.18 | 29.11 ±3.26 | 2.72 ±1.14 | 2.93 ±1.24 | Hysteroscopic Hydrotubation +TCM retention enema | Hysteroscopic Hydrotubation | 3 months | 6 months | NM | NM | Pregnancy rate, Clinical total effective rate |
| Lin and Peng [16] | 35/33 | 28.72 ±1.35 | 29.03 ±1.39 | 3.52 ±1.06 | 3.60 ±1.00 | Conventional Hydrotubation +TCM retention enema | Conventional Hydrotubation | NM | 2 years | more than 1 hour | 15-20cm | Pregnancy rate, Clinical total effective rate, |
| Gao et al. [17] | 20/20 | 29.66 ±4.18 | 29.33 ±3.00 | 2.53 ±0.64 | 2.43 ±0.89 | Conventional Hydrotubation +TCM retention enema | Conventional Hydrotubation | 3 menstrual cycles | 1 year | NM | NM | Pregnancy rate, Clinical total effective rate |
| Rao and Yan [18] | 30/30 | NM | NM | 2–9 | | Hysteroscopic Hydrotubation +TCM Retention enema | Hysteroscopic Hydrotubation | 3 menstrual cycles | 18 months | more than 40 minutes | NM | Pregnancy rate, Clinical total effective rate |
| Xie [19] | 50/50 | 25.76 ±3.1 | 25.68 ±2.9 | 5.65 ±1.28 | 6.02 ±1.16 | Interventional Recanalization+TCM retention enema | Interventional Recanalization | NM | 2 years | more than 1 hour | 15-20cm | Pregnancy rate, Clinical total effective rate |
| Yang [20] | 35/35 | 29.4 ±3.5 | 29.2 ±3.7 | 4.5 ±0.5 | 4.3 ±0.6 | Laparoscopy+TCM retention enema | Laparoscopy | NM | 1 year | 30 minutes | NM | Pregnancy rate, Clinical total effective rate |

(*Continued*)

**Table 1.** (Continued)

| References | No. of patients (T/C) | Mean age (year) T | Mean age (year) C | Disease course (year) T | Disease course (year) C | Intervention T | Intervention C | Course | Follow-up time | TCM retention enema' time | Insertio-n depth | Outcomes |
|---|---|---|---|---|---|---|---|---|---|---|---|---|
| Jin and Jin [21] | 38/37 | 28.56 ±3.17 | 29.14 ±3.25 | 3.72 ±1.18 | 3.67 ±1.14 | Hysteroscopic Hydrotubation +TCM retention enema | Hysteroscopic Hydrotubation | 3 menstrual cycles | 6 months | more than 2 hours | 7-10cm | Pregnancy rate, Clinical total effective rate, Incidence of ectopic pregnancy |
| Huang [22] | 41/41 | 31.1 ±3.5 | 29.3 ±3.2 | 1.7 ±0.4 | 1.6 ±0.3 | Laparoscopy+TCM Retention enema | Laparoscopy | 120 days | 6 months | NM | NM | Pregnancy rate in 1 year, Clinical total effective rate |
| Liu [23] | 30/30 | 28.36 ±4.75 | 27.31 ±5.20 | 6.54 ±2.60 | 6.21 ±2.20 | Hysteroscopic Hydrotubation +TCM retention enema | Hysteroscopic Hydrotubation | 1–3 menstrual cycles | 1 year | NM | 10cm | Pregnancy rate in 1 year, Clinical total effective rate |
| Wang et al. [24] | 50/50 | 31.6 ±4.2 | 32.6 ±3.1 | 3.5 ±1.1 | 3.2 ±1.2 | Conventional Hydrotubation +TCM retention enema | Conventional Hydrotubation | 3 menstrual cycles | 2 years | 20–25 minutes | NM | Pregnancy rate in 2 years, Clinical total effective rate |
| Sun [25] | 34/34 | NM | NM | NM | NM | Conventional Hydrotubation +TCM retention enema | Conventional Hydrotubation | 3 menstrual cycles | 1 year | more than 2 hours | 15-20cm | Pregnancy rate, Clinical total effective rate |
| Li et al. [26] | 71/53 | 30.13 ±4.71 | 30.49 ±4.44 | 4.23 ±1.75 | 3.87 ±2.04 | Hysteroscopy Laparoscopy+TCM retention enema | Hysteroscopy Laparoscopy | 60 days | 1 year | 30 minutes | NM | Pregnancy rate, Clinical total effective rate |
| Xu [27] | 40/40 | 30.6 ±4.4 | 31.4 ±2.1 | 2.52 ±0.24 | 2.61 ±0.25 | Conventional Hydrotubation +Retention enema | Conventional Hydrotubation | 3 menstrual cycles | 1 year | more than 2 hours | 15-20cm | Pregnancy rate in 1 year, Clinical total effective rate, Incidence of ectopic pregnancy |
| Liao et al. [28] | 43/43 | 31.0 ±3.0 | 30.7 ±3.2 | 5.8 ±2.3 | 5.5 ±2.5 | Hysteroscopy Laparoscopy +Retention enema | Hysteroscopy Laparoscopy | 45 days | 1 year | more than 4 hours | NM | Pregnancy rate in 1 year, Clinical total effective rate |
| Wu [29] | 70/70 | 30.23 ±5.09 | 29.77 ±4.53 | 2.90 ±2.43 | 2.74 ±2.41 | Laparoscopy +Retention enema | Laparoscopy | 21days | 18 months | NM | 50cm | Pregnancy rate in 1.5years, Clinical total effective rate, Incidence of ectopic pregnancy |
| Bai and Huang [30] | 40/40 | 29.7 ±1.2 | 30.1 ±2.32 | 5.2 ±0.11 | 4.9 ±0.009 | Conventional Hydrotubation +Retention enema | Conventional Hydrotubation | 45 days | 1 year | NM | NM | Pregnancy rate in 1 year, Clinical total effective rate |
| Zhou et al. [31] | 52/52 | 28.5±1.5 | | 2.0±0.1 | | Laparoscopy +Retention enema | Laparoscopy | 3 menstrual cycles | 1 year | 20-30minutes | NM | Pregnancy rate in 1 year, Clinical total effective rate |

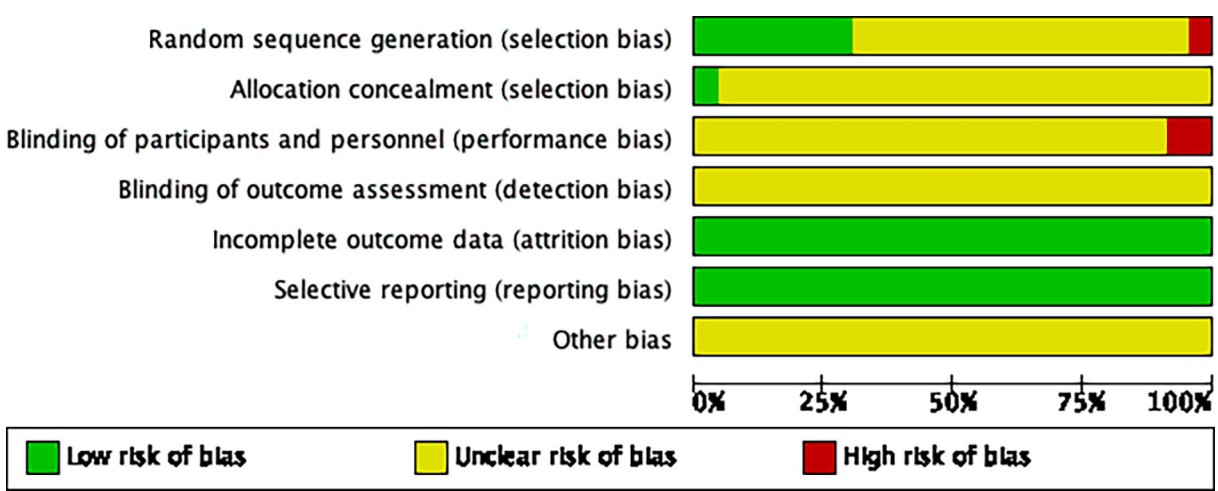

**Fig 2. Risk of bias graph: Authors' judgements about each risk of bias item across all included studies.**

allocation concealment, indicating low risk, and the other 22 studies [9–32] did not show whether they adopted blinding, indicating unclear risk. 2 studies [14, 15] clearly mentioned "patients signed the informed consent forms" but did not mention whether they blinded the test personnel, suggesting high risk. The remaining 21 studies [9–13, 16–31] did not mention it, indicating an unclear risk. The outcome data of all 23 studies [9–31] were complete, or the number of lost cases did not affect the final outcome, suggesting a low risk. All 23 studies [9–31] were not selective reports, indicating a low risk. It could not be determined whether there was other bias in all 23 references [9–31], suggesting an unclear risk (Figs 2 and 3).

## Results comparison

**Clinical pregnancy rate.** Twenty-three trials [9–31] quantified the clinical pregnancy rate. Based on the results of the heterogeneity test ($P = 0.39 > 0.1$, $I^2 = 4.95\%$), the fixed effect analysis model was used to complete the meta-analysis. The meta-analysis showed a higher pregnancy rate in the experimental group than in the control group (RR 1.75, 95% CI [1.58, 1.94], Z = 10.55, $P < 0.00001$; Fig 4).

Although no statistical heterogeneity was observed among the studies, certain clinical heterogeneities might have been present due to surgical options, treatment time, insertion depth of anal canal,follow-up time and TCM retention enema'time. Therefore, the influences of surgical options, treatment time, insertion depth of anal canal, follow-up time and TCM retention enema'time on the clinical pregnancy rate in the five subgroups were compared. Firstly, subgroup analysis showed that TCM retention enema could improve the clinical pregnancy rate whether it was administered common uterine tubal fluid, laparoscope, hysteroscopy or hysteroscopy laparoscopy, respectively. Maybe because hysteroscopy laparoscopy is complexed and multiplex, the included studies had high heterogeneity ($P = 0.04$, $I^2 = 67.88\%$). Secondly,the treatment time was <3 months and≥ three months showed a higher pregnancy rate in the experimental group than in the control group. A relatively long treatment time showed a better intervention effect than a short treatment period.Thirdly, the follow-up time were≤6 months and>6 months showed a higher pregnancy rate in the experimental group than in the control group, and a relatively short follow-up time showed a better intervention effect than a long follow-up period. The fourth, the studies demonstrated that TCM retention enema'time were ≤ 30 minutes and TCM retention enema'time>30 minutes showed a higher pregnancy

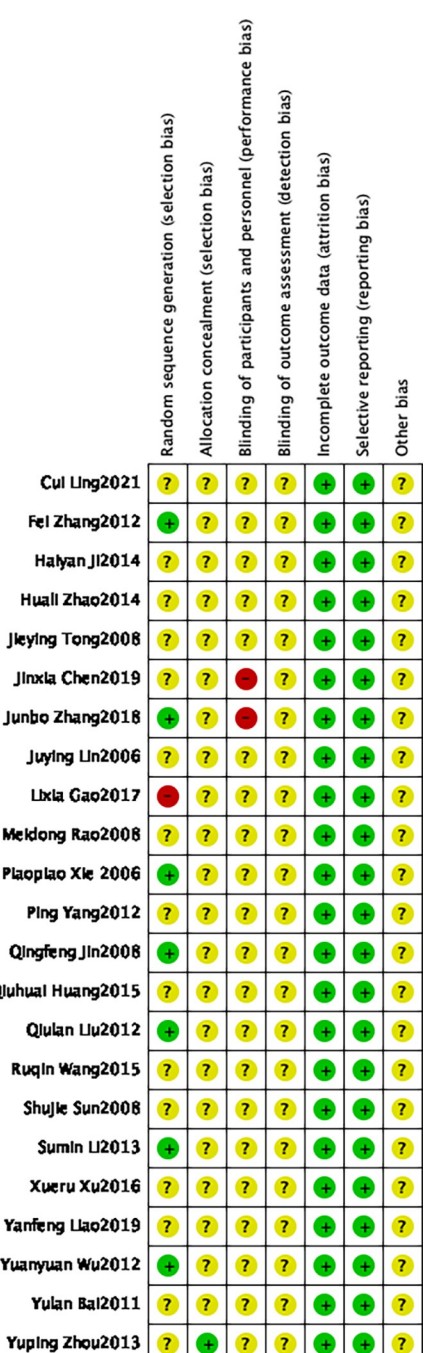

**Fig 3. Risk of bias summary: Authors' judgements about each risk of bias item for each included study.**

rate in the experimental group than in the control group, and a relatively long TCM retention enema'time showed a better intervention effect than a short TCM retention enema'time. Last, the studies demonstrated that insertion depth of anal canal≥15cm and insertion depth of anal canal<15cm a higher pregnancy rate in the experimental group than in the control group, and a relatively short insertion depth of anal canal showed a better intervention effect than a long insertion depth of anal canal (Table 2).

| Study | Treatment Yes | No | Control Yes | No | | Risk ratio with 95% CI | Weight (%) |
|---|---|---|---|---|---|---|---|
| Cui Ling 2021 | 12 | 22 | 5 | 29 | | 2.40 [ 0.95, 6.07] | 1.66 |
| Fei Zhang 2012 | 20 | 22 | 9 | 32 | | 2.17 [ 1.12, 4.19] | 3.03 |
| Haiyan Ji 2014 | 31 | 4 | 20 | 15 | | 1.55 [ 1.14, 2.11] | 6.65 |
| Huali Zhao 2014 | 18 | 42 | 11 | 47 | | 1.58 [ 0.82, 3.05] | 3.72 |
| Jieying Tong 2008 | 28 | 19 | 22 | 24 | | 1.25 [ 0.85, 1.83] | 7.39 |
| Jinxia Chen 2019 | 14 | 10 | 3 | 21 | | 4.67 [ 1.54, 14.18] | 1.00 |
| Junbo Zhang 2018 | 23 | 23 | 12 | 34 | | 1.92 [ 1.09, 3.37] | 3.99 |
| Juying Lin 2006 | 25 | 10 | 17 | 16 | | 1.39 [ 0.94, 2.05] | 5.82 |
| Lixia Gao 2017 | 9 | 11 | 2 | 18 | | 4.50 [ 1.11, 18.27] | 0.66 |
| Meidong Rao 2008 | 19 | 11 | 9 | 21 | | 2.11 [ 1.15, 3.89] | 2.99 |
| Piaopiao Xie 2006 | 43 | 7 | 26 | 24 | | 1.65 [ 1.24, 2.21] | 8.64 |
| Ping Yang 2012 | 22 | 13 | 13 | 22 | | 1.69 [ 1.03, 2.79] | 4.32 |
| Qingfeng Jin 2021 | 21 | 17 | 9 | 28 | | 2.27 [ 1.20, 4.29] | 3.03 |
| Qiuhuai Huang 2015 | 17 | 24 | 8 | 33 | | 2.13 [ 1.03, 4.37] | 2.66 |
| Qiulan Liu 2012 | 8 | 22 | 2 | 28 | | 4.00 [ 0.92, 17.30] | 0.66 |
| Ronghua Sun 2016 | 7 | 24 | 3 | 27 | | 2.26 [ 0.64, 7.93] | 1.01 |
| Shujie Sun 2008 | 16 | 18 | 5 | 29 | | 3.20 [ 1.32, 7.75] | 1.66 |
| Sumin Li 2013 | 38 | 33 | 25 | 28 | | 1.13 [ 0.79, 1.62] | 9.52 |
| Xueru Xu 2016 | 27 | 13 | 15 | 25 | | 1.80 [ 1.14, 2.83] | 4.99 |
| Yanfeng Liao 2019 | 27 | 16 | 18 | 25 | | 1.50 [ 0.98, 2.28] | 5.98 |
| Yuanyuan Wu 2012 | 38 | 32 | 20 | 50 | | 1.90 [ 1.24, 2.92] | 6.65 |
| Yulan Bai 2011 | 25 | 15 | 12 | 28 | | 2.08 [ 1.23, 3.54] | 3.99 |
| Yuping Zhou 2013 | 46 | 6 | 30 | 22 | | 1.53 [ 1.19, 1.97] | 9.97 |
| **Overall** | | | | | | 1.75 [ 1.58, 1.94] | |

Heterogeneity: $I^2$ = 4.95%, $H^2$ = 1.05

Test of $\theta_i = \theta_j$: Q(22) = 23.15, p = 0.39

Test of $\theta = 0$: z = 10.55, p = 0.00

1  2  4  8  16

Fixed-effects Mantel–Haenszel model

**Fig 4. Forest plot: Clinical pregnancy rate in conventional surgery combined with TCM retention enema.** Weights are from fixed-effect analysis.

**Table 2. Subgroup analysis of clinical pregnancy rate for different surgical options, treatment time, insertion depth of anal canal, follow-up time and TCM retention enema time.**

| Subgroups | Number of studies | RR (95% CI) | Z | P | Heterogeneity | |
|---|---|---|---|---|---|---|
| | | | | | $I^2$ | P |
| **Surgical options** | | | | | | |
| hysteroscopy laparoscopy | 3 | 1.59[0.94,2.69] | 1.73 | 0.08 | 66.2% | 0.05 |
| hysteroscopy | 5 | 2.21[1.59,3.07] | 4.71 | <0.00001 | 0% | 0.92 |
| Laparoscope | 5 | 1.62[1.35,1.94] | 5.2 | <0.00001 | 0% | 0.54 |
| Conventional Hydrotubation | 10 | 1.83[1.56,2.15] | 7.44 | <0.00001 | 0% | 0.57 |
| **Treatment time** | | | | | | |
| <3months | 9 | 1.68[1.45,1.94] | 6.88 | <0.00001 | 21.38% | 0.61 |
| ≥3months | 14 | 1.81[1.57,2.10] | 8 | <0.00001 | 0% | 0.22 |
| **Follow-up time** | | | | | | |
| ≤6 months | 5 | 1.97[1.44,2.68] | 4.29 | <0.00001 | 0% | 0.95 |
| >6 months | 18 | 1.71[1.54,1.91] | 9.65 | <0.00001 | 18.49% | 0.23 |
| **TCM retention enema'time** | | | | | | |
| ≤30 minutes | 4 | 1.43[1.21,1.7] | 4.22 | <0.00001 | 0.00% | 0.46 |
| >30 minutes | 10 | 1.87[1.59,2.21] | 7.39 | <0.00001 | 0.00% | 0.59 |
| **Insertion depth** | | | | | | |
| <15cm | 2 | 2.58[1.43,4.65] | 3.16 | <0.00001 | 0.00% | 0.48 |
| ≥15cm | 7 | 1.84[1.54,2.21] | 6.64 | <0.00001 | 0.00% | 0.59 |

**Clinical total effective rate.** Twenty-three studies [9–31] quantified the clinical total effective rate. Based on the results of the heterogeneity test ($P$ = 0.14>0.1, $I^2$ = 25.03%), the fixed effect analysis model was used to complete the meta-analysis. The meta-analysis showed that the clinical total effective rate in the experimental group was higher than that in the control group (RR 1.28, 95% CI [1.28, 1.34], Z = 11.07, $P$<0.00001; Fig 5).

**Incidence of ectopic pregnancy.** Three trials [21, 27, 29] quantified the incidence of ectopic pregnancy. Based on the results of the heterogeneity test ($P$ = 0.97>0.1, $I^2$ = 0.00%), the fixed effect analysis model was used to complete the meta-analysis. The meta-analysis showed that the incidence of ectopic pregnancy in the experimental group was lower than that in the control group (RR 0.40, 95% CI [0.20, 0.77], Z = -2.73, $P$ = 0.01; Fig 6).

**Side effects.** 2 trials reported adverse events of any cause. Xu [27] reported 1 case of distention pain of the lower abdomen, 1 case of nausea and vomiting, 2 cases of vaginal bleeding and 1 case of inflammatory infection in the intervention group, and 2 cases of distention pain of the lower abdomen, 3 cases of nausea and vomiting, 3 cases of vaginal bleeding and 2 cases of inflammatory infection in the control group. Wu [29] reported 3 cases of perianal oedema and 8 cases of increased times in defecation in the intervention group.

**Sensitivity analysis.** The sensitivity analysis of the main outcomes comprising clinical pregnancy rate and clinical total effective rate suggested that removing any one study of each outcome had no significant effect on the overall results, indicating that the results of this meta-analysis were reliable.

**Publication bias.** Funnel plots were made for the pregnancy rate of the main outcome measures, and the results of the funnel plots showed that the left and right distribution of the literature was asymmetric, suggesting that there may have been publication bias (Fig 7). Therefore, we applied a trim and fill analysis in the fixed-effects model (Fig 8) by adding 8 articles; the corrected RR was 1.569, 95% CI: (1.428, 1.725), which represented a reduction from the original effect size of 0.081; however, the change was statistically significant. The results of this study were concluded to be slightly misleading yet acceptable.

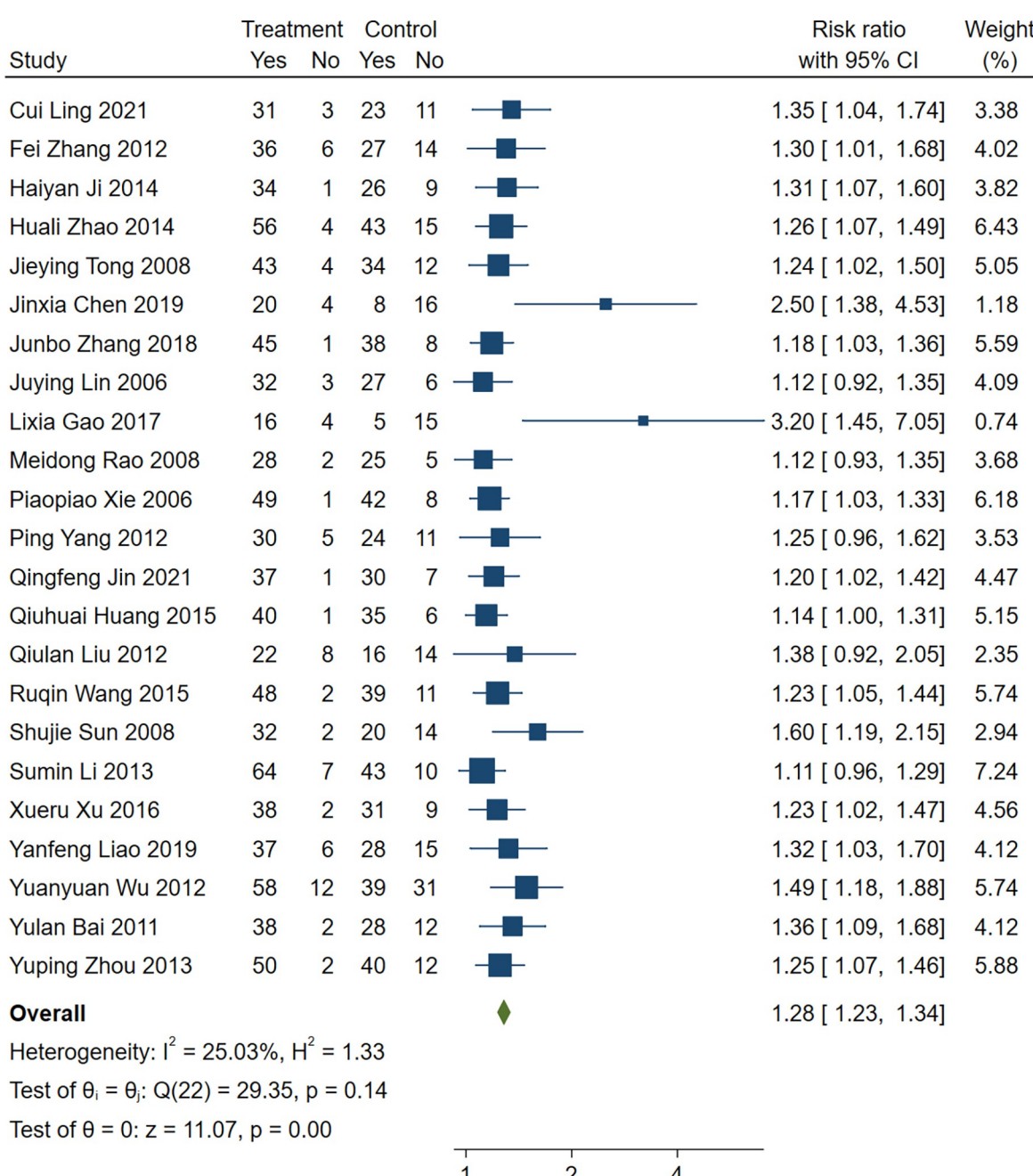

Fig 5. Forest plot: Clinical total effective rate in conventional surgery combined with TCM retention enema. Weights are from fixed-effect analysis.

## Discussion

### Mechanism and effect of TCM retention enema for tubal obstructive infertility

The incidence of infertility is extremely high among women of childbearing age and has been increasing gradually year by year. This gives patients and families significant financial and

**Fig 6. Forest plot: Incidence of ectopic pregnancy in conventional surgery combined with TCM retention enema.** Weights are from fixed-effect analysis.

psychological stress [32]. Tubal infertility is one of the main causes of female infertility. With the progress of medical technology, especially hysteroscopy and laparoscopy, a minimally invasive operation has greatly increased the recanalization rate of blocked fallopian tubes. However, the pregnancy rate is still low and has a strong possibility of ectopic pregnancy [33, 34]. Therefore, we need an auxiliary treatment aimed at improving the efficacy, reducing side effects and improving the prognosis. Conventional surgery combined with traditional Chinese medicinal retention enema is a therapy that has recently been used widely in clinics.

There are many causes of tubal infertility. Among them, pelvic adhesions caused by pelvic inflammatory diseases (PIDs) are the main cause of tubal infertility. Inflammatory reactions are one of the important mechanisms that cause tubal infertility. When tubal inflammation occurs, it can lead to oedema, congestion, hardening and thickening of the fallopian tube. This affects patency and activity of the fallopian tube and prevents sperm and ovum from combining with each other and causes infertility [35]. At the molecular biology level, a variety of cell factors are involved in tubal infertility, including epidermal growth factor (EGF), tumour necrosis factor-alpha (TNF-α), interleukin-2 (IL-2), interleukin-6 (IL-6), interleukin-8 (IL-8) and interferon (IFN-γ) [36]. Several possible hypotheses based on available evidence can explain why traditional Chinese medicinal retention enemas have good efficacy in treating tubal obstructive infertility. The effectiveness of traditional Chinese medicinal retention enemas for tubal obstructive infertility may be attributed to anti-inflammatory properties of TCM and improvement of local blood circulation. The 10 most frequently used Chinese medicines in the included studies were Rhizoma Curcumae, Rhizoma Sparganii, Radix Paeoniae Rubra, Angelica sinensis, Peach Kernel, Sargentodoxa cuneata, Herba Patriniae, Rhizoma Chuanxiong, Fructus Liquidambaris, and Salviae miltiorrhiza. In animal experiments, Rhizoma Curcumae and Rhizoma Sparganii can inhibit fibroblast activation [37]. Radix Paeoniae Rubra, Angelica sinensis, Peach Kernel, Rhizoma Chuanxiong and Salviae miltiorrhizae are blood-activating and stasis-eliminating drugs that have prominent effects on improving blood haemorheology, and preventing platelet aggregation and inflammation [38–41]. Sargentodoxa cuneata and Herba Patriniae have been shown to strongly inhibit inflammation through distinct mechanisms [42]. The main components of Fructus liquidambaris are betulinic acid (also known as liquidambaric acid) and gallic acid [43]. Many studies have confirmed that betulinic

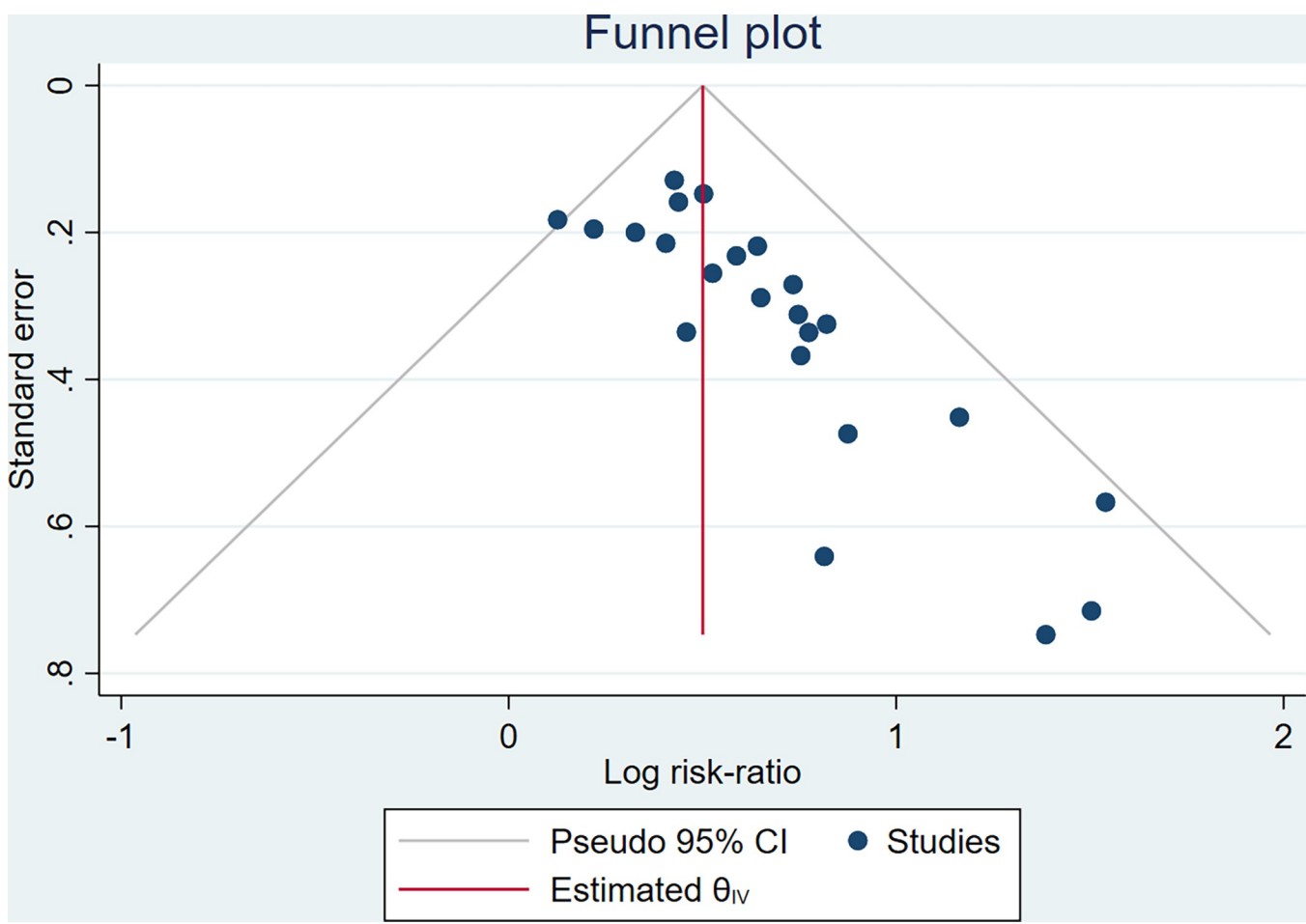

**Fig 7. Funnel plot to evaluate publication bias.**

acid and gallic acid have an anti-inflammatory effect [44, 45]. An animal experiment showed that interventional recanalization combined with retention enema with TCM significantly reduced the secretion of IL-1β, IL-6 and TNF-α and increased the expression of IL-10, which showed that a combination of Chinese and Western medicine can suppress tubal inflammation and prevent tubal adhesion [28]. The other two animal experiments supported these results and suggested that retention enema with TCM can significantly suppress inflammation, improve microcirculation, reduce inflammatory cell infiltration, tissue adhesion and fibrosis, and promote damaged tissue repair [29, 30].

Traditional Chinese medicine enema is a therapy that pours traditional Chinese herbs decoction into the rectum. Anatomically, the fallopian tube is located adjacent to the rectum. Herb decoction can be absorbed directly into the diseased tissue through intestinal mucosa, which can elevate drug concentration quickly and effectively, loosen tissue adhesion, improve local blood circulation, promote resolution of inflammation and accelerate tubal recanalization because the rectal mucosa is rich in blood vessels and the vein wall is thin. Mechanical hydraulic action during enema could also treat slight adhesion of fallopian tubes [46]. Compared with oral administration of decoction, this treatment can reduce the stimulation of drugs to the digestive tract and avoid damage to digestive enzymes. Furthermore, traditional Chinese medicine enema can improve the bioavailability of active components and limit

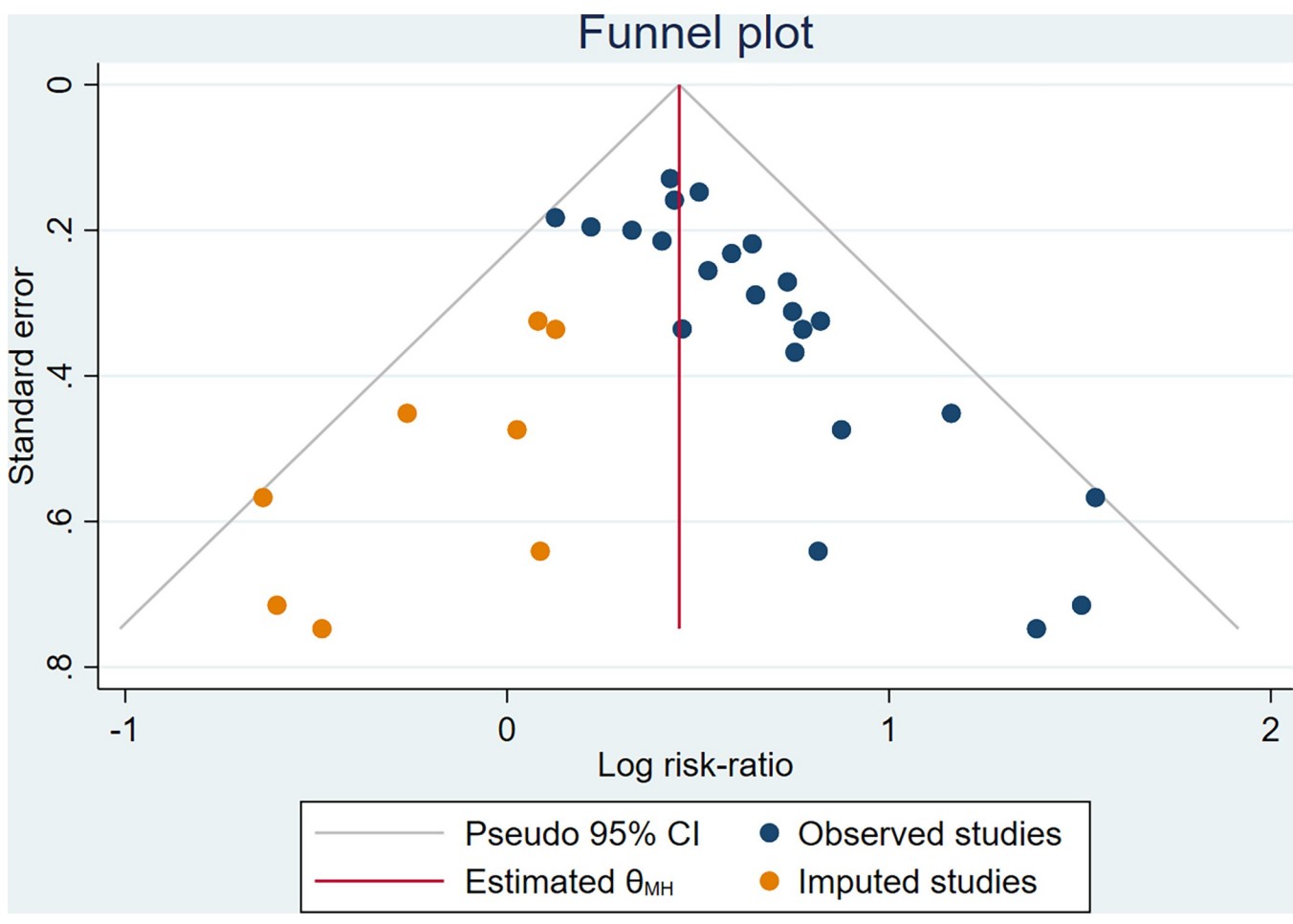

**Fig 8. Trim and fill analysis.**

damage to liver or other organs [47]. Some clinical studies proved that enema administration has a better clinical effect than oral administration in the treatment of tubal infertility [48, 49]. A rat experiment suggested that the oviduct structure in rats treated by clysis improved much better than the rats treated by gavage, and the expression of TNF-α in the clysis group were lower than that in the gavage group [46].

## Summary of main results

In this study, we performed a meta-analysis of 23 trials [9–31] to compare the therapeutic effect of conventional surgery combined with traditional Chinese medicinal retention enema for tubal obstructive infertility in 4547 patients. This meta-analysis showed that conventional surgery combined with traditional Chinese medicinal retention enema can significantly improve the clinical pregnancy rate, improve the clinical total effective rate, lower the incidence of ectopic pregnancy, improve TCM symptoms and improve the signs of obstructive tubal infertility.

According to the subgroup analysis, the studies demonstrated whether the surgical options was common uterine tubal fluid, laparoscope, hysteroscopy or hysteroscopy laparoscopy, showed a higher pregnancy rate in the experimental group than in the control group. The

patients showed the shorter the follow up time,the longer the TCM retention enema'time and the shorter the insertion depth of anal canal showed a higher pregnancy rate.

In addition, only 2 trials [27, 29] reported a total of2adverse events, including diarrhoea, abdominal pain, nausea and vomiting, vaginal bleeding, inflammatory infection, perianal oedema, and increased times in defecation. Sensitivity analysis suggested that these pooled results were stable. There was no obvious heterogeneity in most of the pooling analyses. The funnel plot results reflected that there was an obvious publication bias, but it could be seen from the trim-and-fill analysis that the existing publication bias did not affect the correctness of the conclusion. In summary, traditional Chinese medicinal retention enema is effective in the treatment of tubal obstructive infertility.

### Different conclusions of the published literature

In comparison to prior systematic reviews that included Chinese medicine oral administration and Chinese medicine enema in the treatment group [50], this review focused only on RCTs investigating conventional surgery combined with traditional Chinese medicinal retention enema for tubal obstructive infertility.

### Limitations

This study has several limitations. First, we only searched for literature from Chinese and English databases. All of the trials included in our study were conducted in China. Second, the included trials had generally low methodological quality, and only 7 studies [10, 15, 19, 21, 23, 26, 29] clearly described the use of randomization methods. In most included trials, allocation concealment and blinding were "unclear", which will affect the authenticity of the research conclusions. Third, in this meta-analysis, most Chinese herbal enemas involved had great differences in amounts and compositions, so it posed challenges to explore the differences in efficacy between prescriptions. The times, frequencies and courses of retention enema were different, so we were unable to make a systematic summary.

### Conclusion

This meta-analysis suggested that conventional surgery combined with traditional Chinese medicinal retention enema for tubal obstructive infertility was superior to conventional surgery alone in improving the clinical pregnancy rate, clinical total effective rate and TCM symptoms and lowering the incidence of ectopic pregnancy. Traditional Chinese medicinal retention enema could be recommended as an effective and safe complementary therapy for the treatment of tubal obstructive infertility. However, owing to the small sample size and poor methodological quality of the included studies, long-term and higher-quality RCTs are needed to further investigate the effectiveness and safety of traditional Chinese medicinal retention enemas for tubal obstructive infertility.

### Supporting information

**S1 Checklist. PRISMA 2020 checklist.**
(DOCX)

**S1 Data.**
(XLSX)

## Author Contributions

**Conceptualization:** Sijia Xu, Ling Wang.

**Data curation:** Sijia Xu, Shuo Jin.

**Formal analysis:** Sijia Xu, Shuo Jin.

**Investigation:** Sijia Xu, Liuqing Yang.

**Software:** Sijia Xu, Shuo Jin.

**Supervision:** Sijia Xu, Qin Zhang.

**Writing – original draft:** Sijia Xu, Shuo Jin.

**Writing – review & editing:** Sijia Xu, Qin Zhang.

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
