## [Decision Letter · Decision Letter 0]

30 Jan 2023

PONE-D-22-35231Evidence-based Complementary and Alternative Medicine Conventional Surgery Combined with Traditional Chinese Medicinal Retention Enema for Tubal Obstructive Infertility: A Systematic Review and Meta-analysisPLOS ONE

Dear Dr. Zhang,

Thank you for submitting your manuscript to PLOS ONE. After careful consideration, we feel that it has merit but does not fully meet PLOS ONE’s publication criteria as it currently stands. Therefore, we invite you to submit a revised version of the manuscript that addresses the points raised during the review process.

We look forward to receiving your revised manuscript.

Kind regards,

Ahmed Mohamed Maged, MD

Academic Editor

PLOS ONE

Journal Requirements:

3. Please ensure that you refer to Figure 10 in your text as, if accepted, production will need this reference to link the reader to the figure.

4. Please upload a new copy of Figure 3 and 4  as the detail is not clear. Please follow the link for more information:

https://blogs.plos.org/plos/2019/06/looking-good-tips-for-creating-your-plos-figures-graphics/

https://blogs.plos.org/plos/2019/06/looking-good-tips-for-creating-your-plos-figures-graphics/

**Additional Editor Comments:**

Please respond to all reviewers comments point by point.

Reviewers' comments:

Reviewer's Responses to Questions

**Comments to the Author**

1. Is the manuscript technically sound, and do the data support the conclusions?

Reviewer #1: Yes

Reviewer #2: Partly

2. Has the statistical analysis been performed appropriately and rigorously? 

Reviewer #1: Yes

Reviewer #2: I Don't Know

3. Have the authors made all data underlying the findings in their manuscript fully available?

Reviewer #1: Yes

Reviewer #2: No

4. Is the manuscript presented in an intelligible fashion and written in standard English?

Reviewer #1: Yes

Reviewer #2: No

5. Review Comments to the Author

Reviewer #1: 1- Concerning abstract section,

a-materials and methods, pls widen your research period till November 2022.

b- TCM stands for what? Pls define the abbreviation.

c-results, RCTs stands for, pls define

d- in all manuscript, p value should be written in italics.

2- Introduction

a- What is tubal infertility, pls define it ?

b- The introduction is very superficial, pls add some notes about prevalence and incidence of tubal infertility in China and worldwide.

3- Concerning inclusion and exclusion criteria, are all the studies have the same inclusion and exclusion criteria to be involved in this meta-analysis? And are there any similarities or differences between these studies?

4- Concerning outcome, the incidence of ectopic pregnancy, are all other factors that affect ectopic pregnancy taken into consideration to be included or excluded, such as sexually transmitted diseases, pelvic surgery, endometriosis.

5- Does The data obtained from all clinical trials ,involved in this review, with the same treatment period? or are there any differences ?. As different duration will affect cure rate.

6- Concerning the outcome, I think all studies selected in this meta-analysis must have the same outcome measure to avoid bias. All the studies must affect

Clinical pregnancy rate, Clinical total effective rate, Incidence of ectopic pregnancy, The Improvement of TCM Symptoms, The improvement of TCM symptoms.

Reviewer #2: This manuscript was designed to investigate Evidence-based Complementary and Alternative Medicine Conventional Surgery Combined with Traditional Chinese Medicinal Retention Enema for Tubal Obstructive Infertility: A Systematic Review and Meta-analysis.

Although, the manuscript is technically sound but it needs English editing.

The introduction provides appropriate background of the topic of this research. The results are presented in a bad format. The authors should tabulate the data in collective tables in a good format.

To make this study more substantial and applicable, the author may wish to add recent references specially in introduction.

6. PLOS authors have the option to publish the peer review history of their article (what does this mean?). If published, this will include your full peer review and any attached files.

Reviewer #1: No

Reviewer #2: No

---

## [Author Response · Author response to Decision Letter 0]

5 Mar 2023

Reviewer #1: 

Comment 1: Concerning abstract section,

 a-materials and methods, pls widen your research period till November 2022.

 b- TCM stands for what? Pls define the abbreviation.

 c-results, RCTs stands for, pls define

 d- in all manuscript, p value should be written in italics.

Response 1: a: Thanks for the suggestion. We have searched other studies and checked them carefully. Unfortunately, we found that they did not meet the criteria for inclusion in the research.

b: We define TCM as a complete healing system that developed in China. We have explained it thoroughly in the manuscript. (Line 63-69, Page5)

c: RCTs means randomized controlled trials. (Line 71, Page5)

d: We have modified all p values in all manuscripts.

Comment 2: Introduction

 a- What is tubal infertility, pls define it ?

 b- The introduction is very superficial, pls add some notes about prevalence and incidence of tubal infertility in China and worldwide.

Response 2: a: We defined Tubal Obstructive Infertility as infertility caused by damage to the function and structure of the fallopian tubes due to pelvic infection, gynecological surgery, and other factors, causing chronic inflammatory damage and tissue fibrosis, resulting in inflammatory changes in the fallopian tubes, with pathological changes such as swelling, exudation, fluid accumulation, thickening, adhesion, stiffness, distortion, occlusion, and inaccessibility, leading to obstruction of sperm-egg union or delivery. (Line 46-55, Page4)

b: We revised the introduction and added some data about the prevalence and incidence of tubal infertility in China and worldwide. The global rate of infertility in women is approximately 15.5%, and the incidence is increasing. Tube obstruction can lead to reduced fertility among affected women, accounting for 25 to ~ 35% of all causes of infertility. In a study with 3018 infertility patients in China, we found that 23.46% of the patients had infertility due to tubal factors, especially in people over 35 years old, which is the leading cause of infertility. (Line 42-46, Page6-7)

Comment 3:Concerning inclusion and exclusion criteria, are all the studies have the same inclusion and exclusion criteria to be involved in this meta-analysis? And are there any similarities or differences between these studies?

Response 3:We have revised the inclusion and exclusion criteria(Line 90-120, Page6-7) and checked the studies in this research have the same inclusion and exclusion criteria. Some of the studies have differences, such as the surgical options, treatment time, follow-up time, retention enema time, and insertion depth. We analyze these differences in subgroup analysis. (Table 2, Page 18-19)

Comment 4: Concerning outcome, the incidence of ectopic pregnancy, are all other factors that affect ectopic pregnancy taken into consideration to be included or excluded, such as sexually transmitted diseases, pelvic surgery, endometriosis.

Response 4：Thanks for the suggestion. We added some exclusion criteria, such as sexually transmitted diseases and previous history of abdominal or pelvic surgery, combined with other gynecological diseases of the uterus and pelvis, such as endometriosis. Those with a combination of severe hemorrhoids, anal fissures, and other contraindications to enemas; patients with more serious medical complications and psychiatric disorders, combined with other infertility factors. (Line 108-117, Page7) Furthermore, we also refresh the studies in our research.

Comment 5: Does The data obtained from all clinical trials ,involved in this review, with the same treatment period? or are there any differences ? As different duration will affect cure rate.

Response 5：Thanks for the suggestion. Some of the studies involved in this review have different treatment periods. Thus we added treatment duration in the subgroup analysis. (Table 2, Page 18-19).

Comment 6:Concerning the outcome, I think all studies selected in this meta-analysis must have the same outcome measure to avoid bias. All the studies must affect Clinical pregnancy rate, Clinical total effective rate, incidence of ectopic pregnancy, The Improvement of TCM Symptoms, The improvement of TCM symptoms.

Response 6：Thanks for the suggestion. We consider the Clinical pregnancy rate and Clinical total effective rate to be significant indicators of efficacy. And the incidence of ectopic pregnancy is one of the manifestations of adverse pregnancy. It is not appropriate to directly use it to reflect the clinical effect. Besides, different studies have different definitions of TCM symptoms, which is why we finally deleted TCM symptoms in our research.

Reviewer #2: 

Comment 1: This manuscript was designed to investigate Evidence-based Complementary and Alternative Medicine Conventional Surgery Combined with Traditional Chinese Medicinal Retention Enema for Tubal Obstructive Infertility: A Systematic Review and Meta-analysis.

Although, the manuscript is technically sound but it needs English editing.

The introduction provides appropriate background of the topic of this research. The results are presented in a bad format. The authors should tabulate the data in collective tables in a good format.

To make this study more substantial and applicable, the author may wish to add recent references specially in introduction.

Response 1: We sincerely appreciate the valuable comments. We have tried our best to polish the language in the revised manuscript. Besides, we adjusted the table format to ensure it can be fully presented (Table 1, Page 11-15). We searched the recently published literature but unfortunately found that they did not meet the criteria for inclusion in the research.

---

## [Decision Letter · Decision Letter 1]

3 May 2023

Evidence-based Complementary and Alternative Medicine Conventional Surgery Combined with Traditional Chinese Medicinal Retention Enema for Tubal Obstructive Infertility: A Systematic Review and Meta-analysis

PONE-D-22-35231R1

Dear Dr. Zhang,

We’re pleased to inform you that your manuscript has been judged scientifically suitable for publication and will be formally accepted for publication once it meets all outstanding technical requirements.

Kind regards,

Ahmed Mohamed Maged, MD

Academic Editor

PLOS ONE

Additional Editor Comments (optional):

Reviewers' comments:

Reviewer's Responses to Questions

**Comments to the Author**

1. If the authors have adequately addressed your comments raised in a previous round of review and you feel that this manuscript is now acceptable for publication, you may indicate that here to bypass the “Comments to the Author” section, enter your conflict of interest statement in the “Confidential to Editor” section, and submit your "Accept" recommendation.

Reviewer #1: All comments have been addressed

Reviewer #3: All comments have been addressed

2. Is the manuscript technically sound, and do the data support the conclusions?

Reviewer #1: Yes

Reviewer #3: Yes

3. Has the statistical analysis been performed appropriately and rigorously? 

Reviewer #1: Yes

Reviewer #3: Yes

4. Have the authors made all data underlying the findings in their manuscript fully available?

Reviewer #1: Yes

Reviewer #3: Yes

5. Is the manuscript presented in an intelligible fashion and written in standard English?

Reviewer #1: Yes

Reviewer #3: Yes

6. Review Comments to the Author

Reviewer #1: The revised version of this paper (systematic review and meta analysis ) looks good and the comments were addressed properly.

Reviewer #3: Dear authors: you made a very significant professional topic in this research. Fallopian tubal obstruction to cause infertility is common disease in clinical gynecology. If doing operation unblocking the obstruction can give some help, but it can't make completely heal and recover the function of Fallopian tube, you select TCM retention enema as a complementary treatment with operation for making a higher clinical pregnancy rate, clinical total effective rate, a lower incidence of ectopic pregnancy and improvement of TCM symptoms and signs of obstructive tubal infertility and side-effect. From 50 article of 8 databases, you collect 4547 cases with a standard statistic method for confirming this combined treating model can make superior to conventional operation alone. Whole of your statistic design is very good.

You should compare the herbal prescription in every formula from each article of clinical research for identifying whether there are different treating level in each herbal formula. We should know whether the retention enema is a good treating access or the particular herbal prescription is the key effective substance to make unblocking the obstructive Fallopian tubes. If you can objectively explain this point, your article will have more significance.

7. PLOS authors have the option to publish the peer review history of their article (what does this mean?). If published, this will include your full peer review and any attached files.

Reviewer #1: No

Reviewer #3: **Yes: **Dan Jiang

---

## [Editor Report · Acceptance letter]

9 May 2023

PONE-D-22-35231R1 

Evidence-based complementary and alternative medicine conventional surgery combined with traditional Chinese medicinal retention enema for tubal obstructive infertility: A systematic review and meta-analysis 

Dear Dr. Zhang:

I'm pleased to inform you that your manuscript has been deemed suitable for publication in PLOS ONE. Congratulations! Your manuscript is now with our production department. 

Kind regards, 

on behalf of

Professor Ahmed Mohamed Maged 

Academic Editor

PLOS ONE